# Muscle Quality Index in Morbidly Obesity Patients Related to Metabolic Syndrome Markers and Cardiorespiratory Fitness

**DOI:** 10.3390/nu15112458

**Published:** 2023-05-25

**Authors:** Felipe Caamaño-Navarrete, Daniel Jerez-Mayorga, Cristian Alvarez, Indya del-Cuerpo, Mauricio Cresp-Barría, Pedro Delgado-Floody

**Affiliations:** 1Physical Education Career, Universidad Autónoma de Chile, Temuco 4780000, Chile; marfel77@gmail.com; 2Exercise and Rehabilitation Sciences Institute, School of Physical Therapy, Faculty of Rehabilitation Sciences, Universidad Andres Bello, Santiago 7591538, Chile; daniel.jerez@unab.cl (D.J.-M.); cristian.alvarez@unab.cl (C.A.); 3Strength & Conditioning Laboratory, CTS-642 Research Group, Department of Physical Education and Sports, Faculty of Sport Sciences, University of Granada, 18011 Granada, Spain; indyadelcuerpo@gmail.com; 4Departamento de Educación e Innovación, Facultad de Educación, Universidad Católica de Temuco, Temuco 4780000, Chile; mauriciocrespbarria@gmail.com; 5Department of Physical Education, Sport and Recreation, Universidad de La Frontera, Temuco 4811230, Chile; 6Department of Physical Education and Sports, Faculty of Sport Sciences, University of Granada, 18011 Granada, Spain

**Keywords:** muscle quality index, fitness, severe obesity, metabolic syndrome

## Abstract

Background: Muscle quality index (MQI) is an emerging health indicator obtained by dividing handgrip strength by body mass index (BMI) that needs to be studied in morbidly obese patients (defined by BMI ≥ 35 kg/m^2^). Objective: To determine the association between MQI, metabolic syndrome (MetS) markers, and cardiorespiratory fitness (CRF), and as a second objective to determine the potential mediation role of MQI in the relationship between abdominal obesity and systolic blood pressure (SBP) in this sample. Methods: This cross-sectional study included 86 severely/morbidly obese patients (age = 41.1 ± 11.9 y, nine men). MQI, metabolic syndrome markers, CRF, and anthropometric parameters were measured. Two groups were developed according to MQI; High-MQI (*n* = 41) and Low-MQI (*n* = 45). Results: The Low-MQI group reported higher abdominal obesity (High-MQI: 0.7 ± 0.1 vs. Low-MQI: 0.8 ± 0.1 WC/height; *p* = 0.011), SBP (High-MQI: 133.0 ± 17.5 vs. Low-MQI: 140.1 ± 15.1 mmHg; *p* = 0.048), and lower CRF (High-MQI; 26.3 ± 5.9 vs. Low-MQI; 22.4 ± 6.1 mL/kg/min, *p* = 0.003) than the High-MQI group. Waist-to-height ratio (β: −0.07, *p* = 0.011), SBP (β: −18.47, *p* = 0.001), and CRF (β: 5.21, *p* = 0.011) were linked to MQI. In a mediation model, the indirect effect confirms that MQI is a partial mediator of the association between abdominal obesity with SBP. Conclusions: MQI in morbidly obesity patients reported an inverse association with MetS markers and a positive association with CRF (VO2_max_). It mediates the relationship between abdominal obesity and SBP.

## 1. Introduction

Obesity is a multifactorial disease and has become a worldwide public health problem [1]. Close to a third of the global population is considered overweight or obese [2]—it has almost tripled since 1975 [3]—whereas malnutrition by excess in Latin America is growing quickly compared with other continents [4], which increases the health and economic impact in the countries [5,6]. There is a consensus that obesity may lead to premature disability and death (i.e., cardiovascular diseases (CDV), depression, dementia, and various types of cancers), and therefore, impact life expectancy [7]. Consequently, severe obesity will be the most common body mass index (BMI) classification among women and low-income adults by 2030 in the USA [8]. In this sense, one study reported that the increase in severe obesity was related to the decline of disease-free years in adulthood [9]. This increase in severe obesity could be attributed to the lack of success of public health in managing obesity in its first phases [7].

Similarly, morbidly obesity (defined by the body mass index (BMI) ≥ 40 kg/m^2^ or 35 kg/m^2^ with obesity-related health conditions) [10] has been related to the prevalence of CVD, poor subjective well-being, and bad functional capacity [11]. Moreover, metabolic syndrome (MetS) is a cluster of cardiovascular risk factors including abdominal obesity, high blood pressure, high blood glucose, and blood lipid abnormalities [12]. The MetS prevalence differed from 12.5% to 31.4% in the global population, whereas America and Eastern Mediterranean presented the highest prevalence [13]. Obesity and abdominal obesity have been related to MetS [14,15].

Additionally, the evidence has reported that abdominal obesity is considered a health problem [16,17]. An epidemiological study reported the prevalence of an abdominal obesity increase in adult subjects [18]. Moreover, a systematic review and meta-analysis of 13.3 million participants, indicated a sustained increase in abdominal obesity prevalence since 1990 [19]. Abdominal obesity is a good marker for metabolic disease [20] and may be defined as excess visceral fat in the abdominal region; this condition has reported the association with the development of arterial hypertension [21] and serious implications that promote non-communicable diseases such as heart diseases, non-alcoholic fatty liver diseases, kidney disorders, cancer, and other health problems [22,23].

In a complementary way, abdominal obesity can play a fundamental role in CDV prevention [19]. A recent cross-sectional study found that abdominal fat content was linked with higher blood pressure and arterial stiffness [24]. In this sense, it has been indicated that abdominal obesity was associated with systolic blood pressure and other cardiovascular risk factors [24]. A systematic review of prospective cohort studies reported that the risk of hypertension increased with the elevation of abdominal obesity (i.e., the relative risk of hypertension 1.49 for 10 cm increments in abdominal obesity) [25]. In this sense, a study conducted in adults from urban and rural areas indicated that subjects who had obesity and abdominal obesity obtained a higher risk of hypertension compared to subjects with normal anthropometric measures [26].

Similarly, subjects with abdominal obesity have a significantly higher prevalence of MetS factors involving high systolic blood pressure (SBP), high fasting blood glucose, impaired HDL-cholesterol, and high triglyceride levels [27]. Moreover, the present higher risk of acute myocardial infarction [28] substantially increases the total mortality rates, with most of the excess deaths due to heart disease, and reduces life expectancy when compared with normal weight [29].

On the other hand, handgrip strength (HGS) is an important health indicator and could be easily and universally applied, reporting association with all causes of mortality in different types of populations [30]. HGS is a powerful predictor of future disability, morbidity, and mortality [31]. The evidence suggests that in addition to obesity, a decrease in HGS is associated with an increase in all-cause mortality risk [32]. Moreover, both HGS and obesity evaluated by BMI are predictors of mortality [33]. For this reason, the muscle quality index (MQI) that is obtained by dividing HGS by BMI (MQI = HGS/BMI) is an emerging health and physical function indicator [34,35], which represents relative strength (units; kg/BMI). The evidence suggests that Low-MQI has been associated with all causes of mortality and MetS markers, increasing the risk of CVD, insulin resistance, sarcopenia, and even, death [36,37].

It has been indicated that muscle quality was related to more insulin-sensitive subjects with obesity [38]. Similarly, a recent study found that Low-MQI was associated with a higher risk of diabetes [39]. In this sense, recent evidence indicated that MetS markers such as blood pressure and fasting blood glucose were linked with muscle strength and physical fitness [12]. Yamada et al. [40] found a longitudinal link between low skeletal muscle and the development of MetS. In a complementary way, it has been shown that obese subjects had a high prevalence of poor MQI [41], and poor muscle quality can lead to the development of MetS [42].

In addition, cardiorespiratory fitness (CRF) is a term related to maximal capacity for oxygen consumption and reflected the individual capacity of the circulatory and respiratory system to supply oxygen to the skeletal muscle during physical exercise [43,44,45] and the evidence has reported that it is a strong and independent predictor of CVD [46] and all cause of mortality [44,47]. Moreover, the evidence has shown that CRF is an important health indicator [45]; hence, to present better physical fitness (i.e., CRF and muscular strength), lower MetS markers are reported [48].

A recent longitudinal study indicated that the combination of abdominal obesity with low skeletal mass could be increased the risk of diabetes [49]. Data from severely obese women reported that abdominal obesity was linked with hypertension and other MetS markers [50]. However, the potential mediator role of MQI in the relationship between abdominal obesity and MetS markers has not been studied in deep. Therefore, considering that morbidly obesity patients usually report a major cardiometabolic risk such as elevated MetS markers, the evaluation of the relationship between MQI and different health markers, such as anthropometric, plasmatic, and cardiovascular parameters, is interesting. Therefore, the objective of the present study was to determine the association between MQI with MetS markers and CRF, and as a second objective, to determine the potential mediation role of MQI in the relationship between abdominal obesity and SBP in morbidly obesity patients.

## 2. Materials and Methods

### 2.1. Participants

This cross-sectional study included 86 morbidly obesity patients (age = 41.1 ± 11.9 y, men = 9, BMI; 43.4 ± 7.2). The patients were invited to participate by an open invitation directly from the Morbid Obesity Association (Temuco, Chile) and open information on social networks. After all information and feedback about the risks/benefits were provided, all participants signed an informed consent form. The study was carried out in accordance with the Declaration of Helsinki (2013) and was approved by the Ethical Committee of the Universidad de La Frontera, Temuco, Chile (Act 080-21 and Act 071-18), the database corresponding to project DI21-0030 and DI18-0043.

The inclusion criteria were (i) 18–60 years of age, (ii) to present authorisation medical for physical testing, (iii) BMI ≥ 40 kg/m^2^ (i.e., ≥obesity class III) or BMI ≥ 35 kg/m^2^ with any comorbidity. The exclusion criteria were (i) physical limitations such as restrictive injuries of the musculoskeletal system, (ii) exercise-related dyspnoea or respiratory alterations, and (iii) chronic heart disease with any worsening in the last month. Of the total of participants, eight used drugs for insulin resistance or diabetes, fifteen for hypertension, and three for cholesterol treatment.

### 2.2. Measurements

#### 2.2.1. Muscle Quality Index

The muscle quality index was estimated by HGS divided by BMI. A hydraulic hand dynamometer (BASELINE^®^ Hydraulic Hand Dynamometers, NY, USA) was used to determine HGS, which has been used previously [11]. The measure of each dominant and non-dominant arm was made in two attempts, and the best result from each was selected. The average scores achieved by the left and right hands were registered for data analysis. MQI was categorized as follows: Low-MQI ≤ 50th and High-MQI > 50th (50th MQI = 0.67), according to previous studies [51].

#### 2.2.2. Health Outcomes

To evaluate MetS markers all participants were instructed to arrive at the health centre following overnight fasting >8 h and measured between 08:00 and 09:00 in the morning. The MetS markers considered the following values: fasting plasma glucose (FPG ≥ 100 mg/dL), high-density lipoprotein cholesterol (cHDL < 50 women and <40 mg/dL men), and triglycerides (TG ≥ 150 mg/dL). The additional markers taken were total cholesterol (Tc) and low-density lipoprotein cholesterol (cLDL).

SBP and diastolic blood pressure (DBP) measurements (hypertension = SBP ≥ 140 mmHg, DBP ≥ 90 mmHg) were evaluated by the standard criteria [52]. Blood pressure was measured after 5 min of rest in the sitting position. Two evaluations were made using an OMRON^TM^ digital electronic BP monitor (model HEM 7114, Chicago, IL, USA), and the mean of these measurements was used for data analysis. The participants were informed that they must not drink caffeine or smoke for at least 2 h prior to measurement.

#### 2.2.3. Abdominal Obesity

The waist circumference (WC) of participants was assessed with a non-elastic measuring tape in centimetres (Adult SECA^TM^, USA) at the upper hipbone and top of the right iliac crest, in a horizontal at the level of the iliac crest plane around the abdomen. The tape was snug but did not compress the skin and was parallel to the floor, at the end of a normal expiration the measurement was made. Waist-to-height ratio (WtHR = WC/height) was estimated to determine the central obesity (WtHR ≥ 0.5).

#### 2.2.4. Anthropometric Parameters

The anthropometric variables of participants were made after fasting (>6 h). Body mass (kg) was measured using a digital bio-impedance BIA scale (TANITA^TM^, model 331, Tokyo, Japan), and height (m) was measured using a SECA^TM^ stadiometer (model 214, Hamburg, Germany), with subjects in light clothing and without shoes. The BMI was calculated by dividing body mass in kg by the square of the height in m (kg/m^2^). The BMI was determined to estimate the degree of obesity (kg/m^2^) using the standard criteria for the obesity and severe/morbid-obesity classifications.

#### 2.2.5. Fitness

CRF was measured through the six-minute walking test (6 Mwt). Participants were instructed to walk as far as they could for a 6 min period. An exercise physiologist assisted the participants with instructions. Anthropometric variables and the six-minute walk test results were used to estimate the maximal oxygen uptake (VO2_max_) (mL/kg/min)VO2_max_ from the equation derived by Burr et al. [53]; VO2_max_ = 70.161 + (0.023 × 6 MWT [m]) − (0.276 × weight [kg]) − (6.79 × sex, where m = 0, f = 1) − (0.193 × resting HR [beats per minute]) − (0.191 × age [y]).

### 2.3. Statistical Analysis

The Kolmogorov–Smirnov test was used to determine normal distribution. For continuous variables, values are presented as the mean and standard deviation (SD). Differences between mean values according to MQI were determined using the analysis of variance (ANOVA) and chi-square test (Chi^2^), respectively. A simple linear regression was used to analyse the association between MetS markers and MQI. The alpha level was set at *p* < 0.05 for statistical significance.

Moreover, to verify the effect of the mediating variables MQI (M), considering abdominal obesity (WtHR) as the independent variable (X) and SBP as the dependent variable (Y) regression analyses were performed. Within the analysis, for the samples, the total effect as (c), direct effect as (c′), and indirect effect as (a*b; IE) were calculated, as well as the 95% confidence interval (95%CI), using the macro/interface process v. 3.3 for SPSS v. 23 and the bootstrapping method with a resampling rate of 5000 [54]. The statistical analyses were performed using SPSS statistical software (SPSS^TM^ Inc., Chicago, IL, USA) version 23.0.

## 3. Results

Table 1 shows the comparison according to MQI. In the total sample, the mean age (years), BMI (kg/m^2^), and WC (cm) were 41.1 ± 11.9 years, 43.4 ± 7.2 kg/m^2^ and 121.0 ± 15.2 cm, respectively. The SBP and DBP were 137.7 ± 16.6 mmHg and 86.3 ± 10.3 mmHg, respectively. The mean of mg/dL was 102.4 ± 18.7 for fasting glucose, 185.2 ± 37.4 for total cholesterol, 132.0 ± 64.9 for TG, 115.3 ± 30.6 for cLDL and 48.3 ± 10.9 for cHDL. In fitness, the mean of VO2_max_ (mL/kg/min), HGS (kg), and MQI (ratio) were 24.2 ± 6.3, 31.4 ± 12.3 and 0.7 ± 0.3, respectively. According to MetS factors, the Low-MQI group reported higher abdominal obesity (WtHR, High-MQI: 0.7 ± 0.1 vs. Low-MQI: 0.8 ± 0.1; *p* = 0.011) and SBP (High-MQI: 133.0 ± 17.5 vs. Low-MQI: 140.1 ± 15.1, *p* = 0.048) than the High-MQI group. According to fitness, the Low-MQI reported lower CRF than the High-MQI group (High-MQI; 26.3 ± 5.9 vs. Low-MQI; 22.4 ±6.1 mL/kg/min, *p* = 0.003).

According to MetS markers, there were 47.7% of participants with hypertension, 46.9% reported high fasting glucose, 72% high TG, 39.5% of participants reported low cHDL, and 100% presented abdominal obesity. There were no differences between groups according to MQI (Table 2).

The simple linear regression reported that WtHR (β = 0.07, 95%CI: −0.13, −0.02; *p* = 0.011), SBP (β = −18.47, 95%CI: −28.69, −8.25; *p* = 0.001), and CRF (β = 5.21, 95%CI: 1.21, 9.20; *p* = 0.011) were significantly linked to MQI. When the variables were adjusted by sex and age, the significance was maintained in all (Table 3).

The results of the mediation analysis are shown in Figure 1 for the total sample (*n* = 86). In the relationship between WtHR and SBP, MQI appears as a mediate variable. WtHR was inversely related to MQI (*p* = 0.003) (a) in the first regression step. Second (c), the regression coefficient of WtHR in SBP was also significant (*p* = 0.029). In the third step (b), the potential mediator MQI presented a positive relationship with the dependent variable (Y) (*p* = 0.004), but when both variables in the model WtHR and MQI were included (c′), the regression coefficient remained statistically significant (*p* = 0.029). Finally, the IE confirms that MQI is a partial mediator of SBP (Figure 1).

## 4. Discussion

The objective of the present study was to determine the association between MQI and MetS markers and CRF, and as a second objective, to determine the potential mediation role of MQI in the relationship between abdominal obesity and SBP in morbidly obesity patients. The main results were: (i) MQI was linked inversely to MetS markers (i.e., SBP and abdominal obesity), (ii) the association between abdominal obesity and SBP was mediated by MQI, and (iii) CRF was positively linked to MQI.

MQI was linked inversely to MetS markers (i.e., abdominal obesity) in morbidly obesity participants. In accordance with our results, it has been reported that Low-MQI is associated with a cluster of MetS markers that included SBP and elevated WC [55]. In addition, another study reported that low relative skeletal muscle was associated with a higher risk of MetS [40]. Another retrospective cohort study indicated that low muscle quality and quantity incremented the incidence of MetS [56]. In this sense, it has been shown that low muscle quality, together with visceral fat accumulation, were linked with more prevalence of CVD [57]. Moreover, a study conducted on subjects with overweight or obesity condition indicated that abdominal obesity was related to low muscle quality [58].

In this study, abdominal obesity was linked to SBP, and this relationship was mediated by MQI in morbidly obese patients. In this sense, subjects with abdominal obesity have a significantly higher prevalence of high SBP, high FPG, high Tc, and high TG levels [27]. Additionally, a previous study showed that abdominal obesity was associated with cardiovascular risk factors, low-grade inflammation, SBP, and TG [59]. Another study found that subjects with abdominal obesity presented significantly higher hypertension compared with their counterparts; it has been indicated that abdominal obesity was significantly related to hypertension in a cohort of severely obese women [50] and it has been related to physical fitness impairments such as low CRF and HGS, both limiting the capacity to perform activities of daily living and increasing prevalence of MetS [58,60].

Moreover, it has been indicated that the excess of visceral fat leads to the increased production of adipokines which could have serious implications in non-communicable diseases such as hypertension [22]. Furthermore, a study conducted in 500,000 adult Chinese men and women reported that general adiposity was strongly linked with SBP [61]. According to the mediation role of MQI, the evidence highlights the positive role of muscle fitness in MetS markers [62], where a high muscle fitness could reduce MetS markers (i.e., WC, Tc, cLDL, TG, cHDL, and median blood pressure) [63]; hence, MQI could be a sensitive measure of overall health [64]. MQI is also a great predictor of mortality in both men and women [64]. In addition, low muscle quality has been related to increased CVD risks and all-cause mortality [65] and hypertension [66]. Fortunately, this is something that can be avoided by improving MQI, which significantly changes in response to resistance training [35]. A recent study has indicated that resistance training with both low and moderate loads is just as effective in improving muscular strength and muscle growth as it is in improving MQI [67].

Similar to the above, abdominal obesity was related to MQI and the results of this study agree with those indicated by Palacio-Agüero et al. [68] who reported that subjects with high abdominal obesity presented significantly lower MQI levels using HGS. In this sense, another study conducted in Chilean adults showed that the MQI was negatively linked with abdominal adiposity [69]. A previous study indicated that abdominal obesity had a negative direct effect on HGS in older adults [70]. Similarly, it has been reported that a high body fat percentage is linked to lower muscle quality and predicts a quicker decrease in lean mass [71]. Additionally, overweight and obese women tend to have a lower MQI [72], and a higher adipose tissue is associated with a lower MQI [73]. Moreover, data from 6.455 subjects indicated that HGS was associated with body composition [74]. In addition, evidence has shown that low MQI was associated with CVD in all participants [57]. In an adult population, low values of HGS were linked with high abdominal obesity and the presence of cardiovascular risk factors [75]. In this sense, the increase in the MQI was inversely associated with the risk of MetS in Korean adults [76].

Moreover, MQI was related to CRF (i.e., VO2_max_ in mL/kg/min). An experimental study conducted in diabetics participants reported that changes in muscle quality were linked with improvement in CRF [77]. In addition, a cross-sectional study showed that CRF was positively linked with the skeletal muscle mass index and negatively related to visceral obesity in Korean adults [78]. Similarly, it has been reported that the skeletal muscle mass index was linked to VO2_max_ in older adults [79]. Furthermore, another study conducted in obese and lean sedentary women showed that muscle strength and power were linked with CRF; moreover, the BMI predicted the VO2_max_ [80]. Likewise, another study indicated that better muscle quality was related to higher results in physical function scores in adults [81]. Moreover, it has been shown that there is a link between CRF and muscle quality, either directly or indirectly [81], which agrees with our findings. This is because in the obese population, a higher BMI, which is associated with a lower MQI, leads to a higher percentage of fat mass [82], which in turn leads to low CRF [83]. Excessive fat mass imposes an unfavourable burden on cardiac function and oxygen uptake by the working muscles. This indicates that reduced oxygen utilisation by adipose tissue during exercise reduces the overall CRF [84]. Similarly, the evidence has shown that the population with obesity have severely reduced CRF [85], and muscle mass could be closely linked with CRF [79].

### Strengths and Limitations

The main limitations of the present study are the cross-sectional design. Understanding the mentioned relationships could be better served by the development of a longitudinal study. However, the main strength is novel information on a simple indicator that presents powerful information and relationships with MetS markers. In addition, the sample is small, which prevents the extrapolation of data. For future studies, the role of the MQI should be analysed with more complex methods, such as gold standards (DEXA).

## 5. Conclusions

In conclusion, MQI in morbidly obesity patients reported an inverse association with MetS markers such as abdominal obesity and SBP and a positive association with CRF (VO2_max_). Moreover, it mediates the relationship between abdominal obesity and SBP. Thereby, MQI could be used as a health marker in this population.

## Figures and Tables

**Figure 1 nutrients-15-02458-f001:**
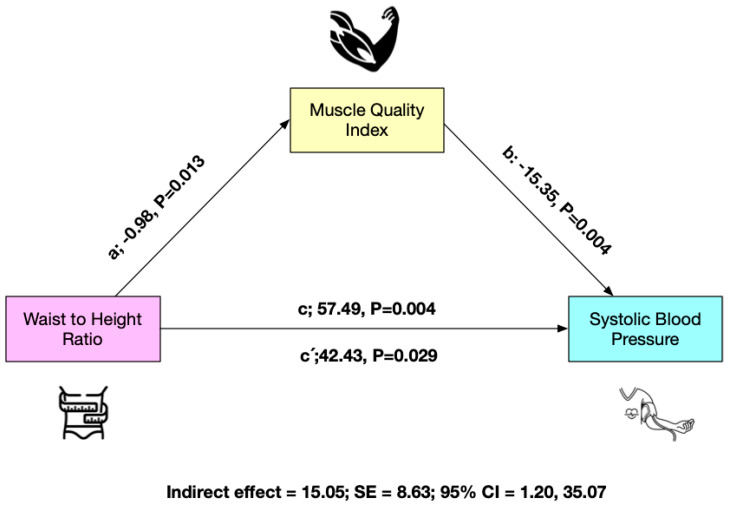
Mediation model testing whether the association between waist-to-height ratio and systolic blood pressure was mediated by muscle quality index.

**Table 1 nutrients-15-02458-t001:** Characteristics of sample study according to muscle quality index.

	All Participants (*n* = 86)	High-MQI (*n* = 41)	Low-MQI (*n* = 45)	
	Mean ± SD	Mean ± SD	Mean ± SD	*p* Value
Age (y)	41.1 ± 11.9	40.2 ± 12.6	42.0 ± 11.4	*p* = 0.492 _(F = 0.48)_
Anthropometrics parameters				
Body mass (kg)	110.7 ± 21.1	108.0 ± 20.6	113.1 ± 21.4	*p* = 0.265 _(F = 1.26)_
Body mass index (kg/m^2^)	43.4 ± 7.2	40.5 ± 5.8	46.0 ± 7.4	*p* < 0.001 _(F = 14.72)_
Waist circumference (cm)	121.0 ± 15.2	119.6 ± 14.0	122.4 ± 16.3	*p* = 0.402 _(F = 0.71)_
WtHR (WC/size)	0.8 ± 0.1	0.7 ± 0.1	0.8 ± 0.1	*p* = 0.011 _(F = 6.73)_
Blood pressure				
Systolic blood pressure (mmHg)	136.7 ± 16.6	133.0 ± 17.5	140.1 ± 15.1	*p* = 0.048 _(F = 4.03)_
Diastolic blood pressure (mmHg)	86.3 ± 10.3	85.1 ± 10.4	87.4 ± 10.2	*p* = 0.308 _(F = 1.05)_
Plasmatic variables				
Fasting Glucose (mg/dL)	102.4 ± 18.7	103.6 ± 21.3	101.4 ± 16.3	*p* = 0.607 _(F = 0.27)_
Total Cholesterol (mg/dL)	185.2 ± 37.4	188.9 ± 36.3	181.9 ± 38.6	*p* = 0.395 _(F = 0.73)_
Triglycerides (mg/dL)	132.0 ± 64.9	127.5 ± 53.8	136.0 ± 74.0	*p* = 0.557 _(F = 0.35)_
cLDL (mg/dL)	115.3 ± 30.6	120.2 ± 28.8	110.7 ± 31.9	*p* = 0.155 _(F = 2.06)_
cHDL (mg/dL)	48.3 ± 10.9	48.2 ± 12.0	48.3 ± 10.0	*p* = 0.969 _(F = 0.00)_
Fitness				
6 Mwt (m)	521.9 ± 116.6	571.0 ± 72.7	477.2 ± 131.0	*p* < 0.001 _(F = 16.38)_
VO2_max_ (mL/kg/min)	24.2 ± 6.3	26.3 ± 5.9	22.4 ± 6.1	*p* = 0.003 _(F = 9.09)_
Handgrip strength (kg)	31.4 ± 12.3	40.6 ± 11.1	23.1 ± 5.4	*p* < 0.001 _(F = 89.61)_
Muscle quality index (ratio)	0.7 ± 0.3	1.0 ± 0.3	0.5 ± 0.1	*p* < 0.001 _(F = 120.96)_

Data are shown in mean and ± standard deviation. Outcomes are described as follows: WtHR; waist-to-height ratio, cHDL; high-density lipoprotein. cLDL; low-density lipoprotein; 6 Mwt; six-minute walking test, MQI; muscle quality index. *p* < 0.05 denotes.

**Table 2 nutrients-15-02458-t002:** Frequency of impaired metabolic syndrome markers according to muscle quality index.

		Total (*n* = 86)	High-MQI (*n* = 41)	Low-MQI (*n* = 45)	*p*-Value
Systolic blood pressure (mmHg) (hypertension)	Normal	45 (52.3%)	24 (58.5%)	21 (46.7%)	*p* = 0.188
	Impaired	41 (47.7%)	17 (41.5%)	24 (53.3%)	
Fasting Glucose (mg/dL) (>100 mg/dL)	Normal	43 (53.1%)	21 (55.3%)	22 (51.2%)	*p* = 0.442
	Impaired	38 (46.9%)	17 (44.7%)	21 (48.8%)	
Triglycerides (mg/dL) (>150 mg/dL).	Normal	59 (72.0%)	26 (66.7%)	33 (76.7%)	*p* = 0.221
	Impaired	23 (28.0%)	13 (33.3%)	10 (23.3%)	
cHDL (mg/dL) (<50 women, <40 men mg/dL),	Normal	34 (39.5%)	17 (41.5%)	17 (37.8%)	*p* = 0.494
	Impaired	52 (60.5%)	24 (58.5%)	28 (62.2%)	
Abdominal obesity (WtHR ≥ 0.5)	Impaired	86 (100.0%)	41 (100.0%)	45 (100.0%)	-

Data are shown as *n* (%). Outcomes are described as follows: cHDL; high-density lipids. WtHR; waist-to-height ratio. *p* < 0.05 denotes significant differences.

**Table 3 nutrients-15-02458-t003:** Association between muscle quality index with MetS markers and physical status.

	β (95% CI)	Beta	SE	t Value	*p*-Value
Waist circumference (cm)	−3.84 (−13.94; 6.27)	−0.08	5.08	−0.76	*p* = 0.452
Adjusted	−11.95 (−21.94; −1.96)	−0.26	5.02	−2.38	*p* = 0.020
WtHR (WC/height)	−0.07 (−0.13; −0.02)	−0.28	0.03	−2.61	*p* = 0.011
Adjusted	−0.10 (−0.17; −0.04)	−0.39	0.03	−3.43	*p* = 0.001
Systolic blood pressure (mmHg)	−18.47 (−28.69; −8.25)	−0.37	5.14	−3.59	*p* = 0.001
Adjusted	−17.89 (−29.47; −6.31)	−0.35	5.82	−3.07	*p* = 0.003
Diastolic blood pressure (mmHg)	−8.21 (−14.79; −1.63)	−0.26	3.31	−2.48	*p* = 0.015
Adjusted	−10.45 (−17.76 −3.15)	−0.33	3.67	−2.85	*p* = 0.006
Fasting Glucose (mg/dL)	2.94 (−10.83; 16.71)	0.05	6.92	0.43	*p* = 0.672
Adjusted	−1.31 (−17.56; 14.94)	−0.02	8.16	−0.16	*p* = 0.873
Total Cholesterol (mg/dL)	10.15 (−14.51; 34.82)	0.09	12.40	0.82	*p* = 0.415
Adjusted	12.01 (−15.02; 39.04)	0.11	13.59	0.88	*p* = 0.379
Triglycerides (mg/dL)	−4.72 (−48.09; 38.65)	−0.02	21.79	−0.22	*p* = 0.829
Adjusted	−19.02 (−67.04; 29.00)	−0.10	24.12	−0.79	*p* = 0.433
cLDL (mg/dL)	6.15 (−14.34; 26.64)	0.07	10.30	0.60	*p* = 0.552
Adjusted	6.43 (−16.90; 29.76)	0.07	11.72	0.55	*p* = 0.585
cHDL(mg/dL)	−1.98 (−9.19; 5.24)	−0.06	3.63	−0.54	*p* = 0.587
Adjusted	1.36 (−6.29; 9.01)	0.04	3.85	0.35	*p* = 0.725
6 MwT	134.70 (63.30; 206.10)	0.38	35.90	3.75	*p* < 0.001
Adjusted	128.61 (47.41; 209.82)	0.36	40.82	3.15	*p* = 0.002
VO2_max_ (mL/kg/min)	5.21 (1.21; 9.20)	0.27	2.00	2.59	*p* = 0.011
Adjusted	8.61 (4.37; 12.85)	0.45	2.13	4.00	*p* < 0.001

Data are shown as β (95% CI) and *p*-values. *p*-values < 0.05 are considered statistically significant. Adjusted represent β adjusted by sex and age. WtHR; waist-to-height ratio, cLDL; low-density lipoprotein, cHDL: high-density lipoprotein, 6 Mwt: six-minute walking test, VO2_max_; Maximal oxygen uptake.

## Data Availability

Not applicable.

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
