# Peer review of "Muscle Quality Index in Morbidly Obesity Patients Related to Metabolic Syndrome Markers and Cardiorespiratory Fitness"

_nutrients, 2023, doi:10.3390/nu15112458_

Round 1

Reviewer 1 Report

The topic considered in this study is an important one and the study has relevance in many domains of interest (in my view it will be of interest to a multi-disciplinary audience).

The pape reads well and, while the manuscript is relatively short, I found that the granularity of the detail provided is adequate to inform the reader of the topic and the results.

I found that the introduction (currently a conftation of an introduction and a brief literature review) requires complete rewriting and revision in two dedicated sections: (a) Introduction: where the background and motivation is considered along with a brief overview of the study, the claimed contribution, and a paper structure, and (b) Related Research; where the studies considered are introduced, discussed, and analysed.

In the study it would be of interest to readers in many domains of interest if there were to be  consideration of open research questions identified in the study.

In summary, I found this to be a generally good paper (revision the current introduction is essential).

Author Response

Reviewer 1

Thank you for the opportunity to improve the paper quality, we think that the article has improved a lot.

The topic considered in this study is an important one and the study has relevance in many domains of interest (in my view it will be of interest to a multi-disciplinary audience).

The pape reads well and, while the manuscript is relatively short, I found that the granularity of the detail provided is adequate to inform the reader of the topic and the results.

Response: Dear reviewer, we have added information according your request

I found that the introduction (currently a conftation of an introduction and a brief literature review) requires complete rewriting and revision in two dedicated sections: (a) Introduction: where the background and motivation is considered along with a brief overview of the study, the claimed contribution, and a paper structure, and (b) Related Research; where the studies considered are introduced, discussed, and analysed.

Response: we have added information in the introduction, you can see in red. 

In the study it would be of interest to readers in many domains of interest if there were to be consideration of open research questions identified in the study.

In summary, I found this to be a generally good paper (revision the current introduction is essential).

Response: We have added your recommendations. Thank you very much for your comments.

Reviewer 2 Report

The article is very well structured, scientifically supported and absolutely sound and logical, though I suggest some changes to improve the impact of the research.

(1)The article needs to address more clearly the novelty of the research. In addition, in the introduction section, I recommend to insert the structure of the article.

(2)In the final section, I think that it would be valuable to enlarge the discussion about the limits of this work, better discussing possible aspects to refine and deepen in future research.

Author Response

Reviewer 2

Thank you for the opportunity to improve the paper quality, we think that the article has improved a lot.

The article is very well structured, scientifically supported and absolutely sound and logical, though I suggest some changes to improve the impact of the research.

(1)The article needs to address more clearly the novelty of the research. In addition, in the introduction section, I recommend to insert the structure of the article.

Response:  We have added information according your recommendations.

(2)In the final section, I think that it would be valuable to enlarge the discussion about the limits of this work, better discussing possible aspects to refine and deepen in future research.

Response:  We have added your recommendations.

Reviewer 3 Report

The study is focused on the muscle quality index in morbidly obese patients and its relation to the markers of metabolic syndrome and cardiorespiratory fitness. The subject is very interesting and it is clearly visible that the authors put a lot of work into conducting the research.
-          A fragment of the template for authors has been left in the first line of the paper

-          Informed Consent Statement section – lines  308, 309 – relates to another article - the authors write about the consent obtained from school-age children, while the average age of the study group in this study exceeds 40 years!

-          The English used requires a lot of corrections, the error appears as early as in the title. The amount of spelling and punctuation errors is beyond acceptation

-          The results are completely unclear and messy

-          Incorrectly conducted and unclear discussion

-          All tables must be consistent and described correctly. Too much information crowding makes the tables illegible

-          References are typical and current, but the format used does not comply with the requirements of the publisher
Taking into account the amount of work put in by the authors in conducting the study I strongly recommend serious rewriting of the paper.

Author Response

Reviewer 3.

Thank you for the opportunity to improve the paper quality, we think that the article has improved a lot.

The study is focused on the muscle quality index in morbidly obese patients and its relation to the markers of metabolic syndrome and cardiorespiratory fitness. The subject is very interesting and it is clearly visible that the authors put a lot of work into conducting the research.

-          A fragment of the template for authors has been left in the first line of the paper

Response: We have corrected.

-          Informed Consent Statement section – lines  308, 309 – relates to another article - the authors write about the consent obtained from school-age children, while the average age of the study group in this study exceeds 40 years!.

Response: We have added your recommendation.

-          The English used requires a lot of corrections, the error appears as early as in the title. The amount of spelling and punctuation errors is beyond acceptation

Response: we have reviewed all text and added more information.

-          The results are completely unclear and messy

Response: We have changed it, according your recommendation. Table 1 was changed.

-          Incorrectly conducted and unclear discussion

Response: We have added information to discussion and corrected some parts.

-          All tables must be consistent and described correctly. Too much information crowding makes the tables illegible

Response: we have adapted according your request.

-          References are typical and current, but the format used does not comply with the requirements of the publisher.

Response: We have changed it.

Taking into account the amount of work put in by the authors in conducting the study I strongly recommend serious rewriting of the paper.

Response: Done, you can see all changes in red and yellow.